# Use of Hypolipidemic Drugs and the Risk of Second Primary Malignancy in Colorectal Cancer Patients

**DOI:** 10.3390/cancers14071699

**Published:** 2022-03-27

**Authors:** Jana Halámková, Lucia Bohovicová, Lucie Pehalová, Roman Goněc, Teodor Staněk, Tomáš Kazda, Lucie Mouková, Dagmar Adámková Krákorová, Šárka Kozáková, Marek Svoboda, Regina Demlová, Igor Kiss

**Affiliations:** 1Department of Comprehensive Cancer Care, Masaryk Memorial Cancer Institute, 656 53 Brno, Czech Republic; jana.halamkova@mou.cz (J.H.); lucia.bohovicova@mou.cz (L.B.); dadamkova@mou.cz (D.A.K.); msvoboda@mou.cz (M.S.); kiss@mou.cz (I.K.); 2Department of Comprehensive Cancer Care, Faculty of Medicine, Masaryk University, 625 00 Brno, Czech Republic; 3Department of Medical Ethics, Faculty of Medicine, Masaryk University, 625 00 Brno, Czech Republic; 4Institute of Health Information and Statistics of the Czech Republic, 128 01 Prague, Czech Republic; pehalova@iba.muni.cz; 5Institute of Biostatistics and Analyses, Faculty of Medicine, Masaryk University, 625 00 Brno, Czech Republic; 6Department of Pharmacy, Masaryk Memorial Cancer Institute, 656 53 Brno, Czech Republic; roman.gonec@mou.cz; 7Department of Surgical Oncology, Masaryk Memorial Cancer Institute, 656 53 Brno, Czech Republic; teodor.stanek@mou.cz (T.S.); moukova@mou.cz (L.M.); 8Department of Surgical Oncology, Faculty of Medicine, Masaryk University, 625 00 Brno, Czech Republic; 9Department of Radiation Oncology, Masaryk Memorial Cancer Institute, 656 53 Brno, Czech Republic; tomas.kazda@mou.cz; 10Department of Radiation Oncology, Faculty of Medicine, Masaryk University, 625 00 Brno, Czech Republic; 11Department of Pharmacy, University Hospital Brno, 625 00 Brno, Czech Republic; kozakova.sarka@fnbrno.cz; 12Department of Pharmacology, Faculty of Medicine, Masaryk University, 625 00 Brno, Czech Republic; 13Clinical Trial Unit, Masaryk Memorial Cancer Institute, 656 53 Brno, Czech Republic

**Keywords:** hypolipidemic agents, statins, second primary malignancies, second primary cancers, multiple primary neoplasms, colorectal cancer, cancer survivors

## Abstract

**Simple Summary:**

Hypolipidemic drugs are among the most frequently prescribed medications in the Western world. Since many studies have indicated their role in carcinogenesis, this work aimed to investigate their association with the occurrence of a second primary malignancy in colorectal cancer survivors. The overall incidence of a second neoplasm was not linked to hypolipidemic medication; however, a subgroup analysis revealed a lower incidence of secondary neoplasia in statin users. When stratified by cancer types, a significant increase in gastric and bladder cancer was detected among colorectal cancer patients using hypolipidemic drugs. Survival outcomes in patients with early-stage colorectal carcinoma who suffered second cancer were significantly worse if treated with hypolipidemic drugs. Although our results do not provide evidence for a causative relationship between hypolipidemic medication and carcinogenesis, these correlations might steer the direction of tertiary prevention care towards specific risk factors shared between cardiovascular diseases and cancer.

**Abstract:**

An increasing number of studies has brought evidence of the protective role of statin use against different types of cancer. However, data on their association with second primary malignancies (SPMs) are lacking. The purpose of this study was to determine the role of hypolipidemic treatment in the prevention of second primary cancer in colorectal cancer (CRC) survivors. We conducted a retrospective single-institution study of 1401 patients with newly diagnosed colorectal cancer from January 2003 to December 2016, with follow-up until December 2020. An SPM was detected in 301 patients (21%), and the incidence was significantly lower in patients with statin medication. However, stratification by cancer types revealed an increased incidence of bladder and gastric cancer in hypolipidemic users. A Kaplan−Meier analysis of early-stage CRC survivors with an SPM showed a significant survival benefit in patients without a history of hypolipidemic treatment. Despite the protective role of statins on overall second cancer incidence, these data indicate that CRC survivors treated with hypolipidemic drugs should be screened more cautiously for SPMs, especially for gastric and bladder cancer.

## 1. Introduction

Colorectal carcinoma (CRC) belongs to the most prevalent cancer diagnoses in Western countries. In the Czech Republic, with more than 10 million inhabitants, the prevalence of people with a history of colorectal cancer reached 59,166 cases in the year 2018 [1].

There is an increasing amount of literature indicating that carcinogenesis may be affected by impairments in cholesterol metabolism. Cholesterol plays crucial roles in various intracellular processes that regulate cell metabolism and maintain homeostasis [2]. Moreover, it is a vitamin D precursor and a substrate for the synthesis of hormones, including estrogens, androgens, progestogens, and corticosteroids that may be involved in the progression of specific cancers [3]. Cholesterol supply, which is necessary for the production of cellular membranes, may also act as a limiting factor for the proliferation of cancer cells.

Statins are lipid-lowering drugs that are widely used in the management of cardio- and cerebrovascular diseases. By inhibiting 3-hydroxy-3-methylglutaryl-coenzyme A (HMG-CoA) reductase, they interfere with cholesterol synthesis through the mevalonate cascade [4]. Accumulating preclinical studies suggest the oncogenic potential of this pathway through either cholesterol-related [5,6] or cholesterol-independent mechanisms [7]. The pleiotropic effects of statins at the cellular level extend beyond their lipid-lowering purpose and include multiple impacts on cell cycle regulation. Statins have been shown to induce p21 (also termed CDKN1A) that plays an important role in cell cycle inhibition [8,9] to antagonize RAS-dependent signaling pathways [10,11] or to downregulate BCL-2 expression in cancer cell lines and, thus, affect apoptosis [12].

Despite the abundance of laboratory evidence of deregulated metabolic pathways and other molecular mechanisms induced by statins, their causative role in cancer chemoprevention has not been unequivocally demonstrated. A retrospective cohort and case−control studies that report reduced cancer-related mortality [13,14,15] or cancer incidence [16] among statin users are often disputed by other studies, showing little impact of statins on cancer-specific mortality [17] or overall cancer incidence [18,19]. These inconclusive results prompted several large meta-analyses, most of which did not support a protective role of statins [20,21,22,23]. Similarly, a meta-analysis of 27 randomized clinical trials (RCTs) did not provide sufficient evidence for an association between statins and aggregate cancer incidence or mortality [24].

The impact of hypolipidemic drugs on the incidence of specific types of cancer has been investigated mainly in breast, prostate, and colorectal cancer patients but less frequently in lung, gastric, and gynecological cancers. Nationwide prospective studies of breast cancer patients have both favored [25,26] and disputed [27] the protective effect of statins. Several meta-analyses of observational studies and RCTs revealed a null association between statin use and the risk of breast cancer development or recurrence [28,29,30]; however, the lists of the original studies involved in these analyses overlap significantly. Contrarily, Liu et al. [31] and Manthravadi et al. [32] reported improved recurrence rates and cancer-specific survival, mainly among lipophilic statin users.

There are several case−control studies suggesting an inverse relationship between statin use and prostate cancer risk [33]; however, the results are limited to hydrophobic statins [34], to certain subgroups of men with concomitant use of NSAIDs [35], or to patients with advanced disease [36]. On the other hand, a number of studies do not support the association between the use of statins and prostate cancer [37,38,39,40,41].

Similarly, there are several large population-based studies that did not demonstrate a reduced risk of CRC incidence related to the use of statins [42,43]. A meta-analysis of 31 observational and randomized studies by Bardou et al. [44] reported a 9% reduction in the incidence of CRC among statin users. Likewise, a meta-analysis of 42 studies by Liu et al. [45] also indicated the protective role of statins on CRC incidence. However, in a subgroup analysis, patients with a history of long-term statin use (more than 5 years) did not experience any CRC risk−benefit.

There is only little information related to cancer and the use of fibrates. In vitro experiments have implicated that fenofibrate might regulate cell cycle, promote apoptosis, or suppress cell proliferation and tissue differentiation by activating PPARA (peroxisome proliferator-activated receptor alpha) [46]. This class of nuclear receptors act on the CDKN2A/RB/E2F pathway that is critical for the control of malignant cell proliferation. In clinical studies, fenofibrates were found to induce apoptosis in triple-negative breast cancer [47], inhibit pancreatic cancer cell proliferation [48], or reduce the risk of hepatocellular carcinoma [49]. Nonetheless, a meta-analysis of 17 relevant RCTs did not demonstrate an impact of fibrates on cancer incidence or mortality [50].

The success rate of CRC treatment is increasing and the prolonged survival of cancer patients increases their probability of developing a second primary malignancy (SPM). Data on the type and frequency of SPMs and potential risk factors are essential for high-quality survivorship care and patient-tailored cancer surveillance. Since the risk factors for CRC and cardiovascular diseases largely overlap (smoking, overweight, little physical activity, etc.), it can be assumed that many CRC patients are treated for hyperlipidemia. Thus, the aim of our study was to ascertain the impact of hypolipidemic drugs on the incidence of second malignancies and overall survival in a cohort of patients with primary CRC.

## 2. Materials and Methods

In this study, we included adult patients with a histologically confirmed CRC diagnosed at the Masaryk Memorial Cancer Institute (MMCI) in Brno, Czech Republic from January 2003 to December 2016. Patients with CRC diagnosed at autopsy, individuals with a high risk of the development of SPMs due to hereditary cancer syndrome (e.g., BRCA1,2, Lynch syndrome, or familial adenomatous polyposis), and patients lost to follow-up were excluded from the study. Enrollees were followed until December 2020. 

Basic clinical data and patient characteristics were retrieved from our institution’s electronic health records. We identified the hypolipidemic drug users based on the medication history whereas the treatment of hyperlipidemia (HLP) had to precede patient’s first cancer diagnosis. Nonusers were defined as patients who have never used a hypolipidemic agent before or during the study period.

Following the National Cancer Institute’s definition, tumors were considered multiple primary malignancies if they differed in topography, histology, or morphology group and were not an extension, a recurrence, or a metastasis [51,52]. Thus, lesions with ICD codes C79.0–C79.9 (secondary malignant neoplasms) were not included in the analysis. All of the SPMs were histologically verified, and their identification is based on the pathologist’s report.

Comparisons of the basic characteristics between patients with an SPM and those without an SPM were summarized with counts and frequencies. Categorical characteristics were tested with the Fisher exact test. The Fisher exact test was also used to assess the relationship between the occurrence of SPMs with the use of hypolipidemic drugs and the laterality of CRC. For continuous characteristics, the median, 25–75% percentile, and Mann−Whitney test were used.

Multivariate logistic regression models were used to adjust risk factors in the analysis (gender, age at CRC diagnosis, clinical stage, status of relapse, and laterality) of the association between an SPM and the use of hypolipidemic drugs. Patients with an unknown clinical stage and a diagnosis of C18.4 were removed from the analysis.

A comparison of the occurrence of SPMs by site in patients with or without hypolipidemic drugs was performed by the N-1 chi-squared test. SPMs with an unknown date of diagnosis were not included in the analysis (7 cases). The Czech National Cancer Registry [1] was used as a reference for the frequencies of selected cancer types in the entire Czech population over the corresponding period of time.

Kaplan−Meier curves were plotted to display the survival of patients with CRC, stratified by the occurrence of an SPM, clinical stage, and use of hypolipidemic drugs. A 15-year survival was used as the primary endpoint. Observations were censored at 15 years of follow-up, with 73 subjects remaining right-censored at this point. Differences in survival between defined groups of patients with respect to the use of hypolipidemic drugs and the occurrence of SPMs were analyzed with the Breslow test.

## 3. Results

We identified a cohort of 1401 patients with primary CRC. The median age was 64 years, and 855 patients were men (61%). Other baseline patient and tumor characteristics stratified by the occurrence of an SPM are listed in Table 1.

The median follow-up was 9.01 years; 723 patients have died during the study period, but none of the patients was lost to follow-up.

One or more SPMs were diagnosed in 301 patients (21.5%), with 55 patients (3.9%) having two or more SPMs. (Table 2).

The incidence of an SPM was significantly higher among older patients and in those with a lower stage CRC, probably reflecting their better overall survival. Hyperlipidemia (HPL) was diagnosed in 257 (18.3%) patients. The analysis did not show any significant difference in the HPL distribution and the type of hypolipidemic treatment between patients with an SPM (15.9%) and without an SPM (19%), as summarized in Table 3.

The use of hypolipidemic agents was associated with a nonsignificant, lower incidence of an SPM (18.7% vs. 22.1% in nonusers), and the anticarcinogenic effect was more pronounced among statin users (17.0% vs. 22.2% in nonusers). Given the low frequency of hydrophilic statin use (7.4% of all hypolipidemic drugs), the association remained significant only for lipophilic agents when statins were categorized by their solubility. Fibrates did not demonstrate a significant chemoprotective effect against cancer although this subgroup of patients was too small (N = 28), and the statistics were underpowered.

After the adjustment for gender, age at CRC diagnosis, clinical stage, status of relapse, and laterality, the odds ratios for the occurrence of an SPM were significantly lower in a subgroup of statin users (Table 4).

As demonstrated in Table 5, the incidence of an SPM was associated with the anatomical distribution of the primary colorectal cancer. In our cohort, patients with an SPM had a higher prevalence of proximal colon cancer (23.6%) and a lower frequency of rectal cancer (44.0%) compared to patients without an SPM (18.4% and 51.8%, respectively). When stratified by hypolipidemic drug use, nonusers with SPMs had a higher proportion of proximal colon cancer (23.9%) compared to patients without any second malignancy (17.7%). However, this correlation between laterality and SPM occurrence was not detected in patients treated for hyperlipidemia (21.3% vs. 21.7%).

Table 6 shows the type and frequency of SPMs in patients taking hypolipidemic agents versus nonusers. The last column (CNCR) serves as a reference, indicating the incidence rate of a particular neoplasm in the Czech Republic over the corresponding period of time.

Figure 1 provides a bar chart for the visual presentation of the data, stratified by the use of hypolipidemic drugs. A statistically significant increase in the prevalence of an SPM among hypolipidemic users was detected in the subgroup of patients with bladder and gastric cancer.

The Kaplan−Meier curves shown in Figure 2 were used to compare the overall survival of CRC patients, stratified by the occurrence of an SPM and the use of hypolipidemic drugs according to the clinical stage of the disease. A statistically significant survival benefit was observed in early-stage CRC patients without an SPM who were not treated with hypolipidemic agents while the worst outcomes were found in patients with SPMs and hypolipidemic treatment. The median overall survival (mOS) of patients who developed a second malignancy was significantly shorter for those using hypolipidemic drugs (6.3 years) compared to nonusers (10.1 years) in early-stage cancer survivors (*p* < 0.001). This indicates that the use of hypolipidemic medication or hyperlipidemia itself might be an unfavorable prognostic factor for overall survival in this subgroup of patients. For patients with stage III and IV CRC, the difference in survival between groups was not statistically significant.

## 4. Discussion

In our cohort of CRC patients, we found little evidence of an anticarcinogenic potential of hypolipidemic treatment. However, subgroup analysis of statins and fibrates revealed a weak protective effect of statins on the occurrence of an SPM, but statistical significance was achieved only after an adjustment for known covariates.

There is a paucity of studies addressing the impact of statins on the development of a second malignancy in cancer survivors, except for Lu et al., who did not demonstrate a significant effect of statins on the frequency of second malignancies in a cohort of breast cancer patients [53]. There are several studies that investigated the effect of statins on cancer-specific mortality or cancer recurrence in colorectal cancer patients. Most favor a protective effect, such as the large-population-based studies of Cardwell et.al. [54] or Voorneveld et al. [55] that reported cancer-specific mortality reduction in patients with a history of postdiagnosis or prediagnosis statin use, respectively. Correspondingly, several meta-analyses suggest that both pre- and postdiagnostic use of statins might improve the cancer-specific survival [56,57,58]. Unfortunately, patients who died from second primary cancers were often not included in the cancer-related mortality subgroup, and data on the type and frequency of second malignancies are not available.

In our study, we found a lower incidence of an SPM among statin users but not in patients treated with hypolipidemic drugs in general. This finding implies a specific protective mechanism inherent to statins rather than the treatment of hyperlipidemia as such. Accordingly, several reports have provided evidence that serum lipid levels do not influence CRC recurrence [59]. On the other hand, when hypothesizing about the anticancer effects of statins, it is of note that the reduction of cancer-specific mortality in statin users might not necessarily be associated with a decrease in CRC recurrence. Based on the findings of reduced CRC-specific mortality but not the CRC recurrence rate, Lash et al. suggest that statins do not have a direct anticancer effect but provide postrecurrence survival benefits by another mechanism [60]. This is supported by the findings of Ng et al., who also reported a null association of statin use with CRC outcomes and recurrence rates in patients with advanced-stage disease [61]. Similarly, the cohort study of Gray et al. [62] did not find any impact of postdiagnosis statin use on cancer-specific survival [62].

Our survival analysis revealed that the occurrence of secondary malignancy or the use of hypolipidemic drugs does not affect overall mortality in advanced-stage colorectal cancer patients. This may indicate that the impact of statins is obscured by a more significant risk factor, such as the advanced stage of the primary cancer itself, the use of chemotherapy, or other covariates that have not been addressed in the analysis. In contrast, our findings in stage I and II colorectal carcinoma show that hypolipidemic medication in combination with an SPM is a significant adverse factor for patient survival. Whether this is attributable to the medication itself or to the diagnosis of hyperlipidemia and other diseases aggregated within the metabolic syndrome remains to be elucidated.

An unexpected finding was the statistically significant increase in gastric and bladder cancer in patients taking hypolipidemic agents. The increased risk of occurrence of bladder cancer in statin users is in accordance with the study of Lundberg et al. [63], who followed a cohort of 22,936 patients with newly detected urothelial bladder cancer. He hypothesized that the tumor-promoting effect of statins might be mediated by mitochondrial changes in urothelial cells caused by metabolic products of statins after their excretion. Most statins are metabolized by the liver and excreted to the bile, except for hydrophilic statins (pravastatin and rosuvastatin), simvastatin, and lovastatin that are typically eliminated by renal clearance [64]. This might be the reason for the idiosyncratic results, indicating a positive relationship between this type of cancer and the use of hypolipidemic agents. A similar association was described by Guercio et al., who also reported a nonsignificant increase in bladder cancer risk among statin users [65] as did the meta-analysis of Zhang et al. [66].

An increased risk of gastric cancer associated with the use of hypolipidemic drugs might be explained by an alteration of the host’s cellular adaptive mechanisms to H. pylori-induced stress, which involves deregulation of the unfolded protein response (UPR) and autophagy [67]. UPR is a protective mechanism activated by endoplasmatic reticulum stress that aims to maintain homeostasis by increasing cellular capacity to process proteins and by inducing autophagy of abundant organelles [68,69]. Statins are thought to stimulate the UPR and, thus, increase cell survival, which might result in the malignant transformation of normal cells or promote an aggressive phenotype of precancerous lesions [70]. Nevertheless, our results contrast with clinical studies that evaluated the impact of statins on the risk of cancer, as most of those reported a chemoprotective potential of these drugs [71,72,73]. Similarly, several meta-analyses of studies involving gastric cancer cases suggested a significant risk reduction in the incidence of gastric cancer related to the use of statins [74,75]. A reasonable explanation for this contrast might, again, lie in confounding by unmeasured variables, such as obesity, which is often associated with hyperlipidemia as a part of the metabolic syndrome, as well as dietary and lifestyle habits, smoking, or other prognostic factors that were not addressed in our analysis.

In a previous analysis, we showed that diabetes mellitus is a negative prognostic factor in CRC patients with an SPM [76], and our current analysis suggests that the use of statins or the treatment of hyperlipidemia itself might also be predictive of cancer patients’ outcomes. To the best of our knowledge, this is the first article evaluating the association between hypolipidemic treatment and the development of SPMs in colorectal cancer patients. The strengths of our study include a long follow-up period of up to 17 years (median follow-up 9.01 years) and the identification of hypolipidemic treatment prior to the diagnosis of the primary cancer, which minimized the impact of immortal time bias [77] on the survival analysis. Our study is based on a large sample and provides reliable information on patient characteristics and the type and frequency of all second malignancies. Due to the retrospective nature of our study, information on the adherence to hypolipidemic treatment and length of statin exposure is missing as are dietary and lifestyle habits, smoking, or other prognostic factors.

## 5. Conclusions

In our analysis, the use of hypolipidemic agents was associated with a lower incidence of an SPM, where the protective effect was most prominent in statin users. Despite this trend, further analysis revealed a reduction of the overall survival in early-stage colorectal cancer patients with an SPM treated by hypolipidemic drugs. Thus, we recommend that CRC patients using hypolipidemic treatment should be regularly screened not only for CRC recurrence but also for SPMs, particularly bladder and gastric cancer. In a previous analysis, we showed that diabetes mellitus is a negative prognostic factor in CRC patients with an SPM. Similarly, our current results suggest that the use of statins or hyperlipidemia itself might also be predictive of cancer patient outcomes. We propose that patients with chronic diseases, such as hyperlipidemia or diabetes mellitus, are candidates for consistent tertiary prevention based on a personalized analysis of associated risk factors.

## Figures and Tables

**Figure 1 cancers-14-01699-f001:**
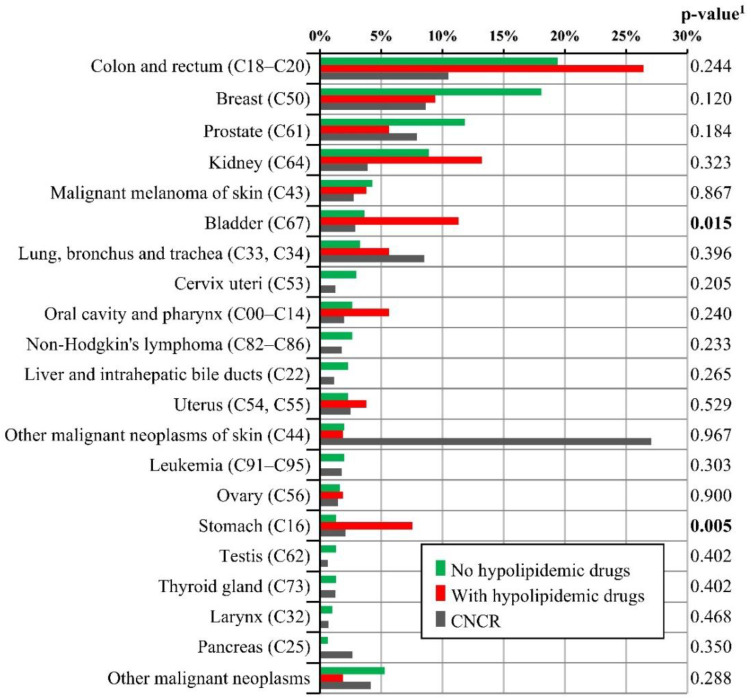
Comparison of the occurrence of second primary malignancies with respect to the use of a hypolipidemic drug. (^1^ *p*-value of N-1 Chi-squared test for the group of nonhypolipidemic users and the group of patients with hypolipidemic drugs. CNCR, Czech National Cancer Registry).

**Figure 2 cancers-14-01699-f002:**
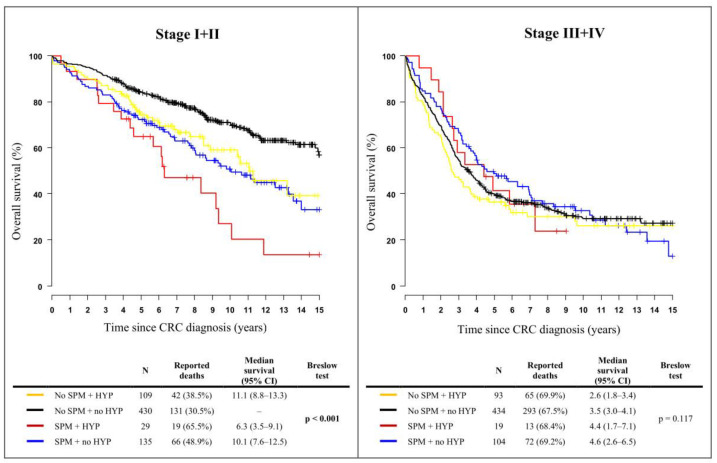
Kaplan−Meier curves of a 15-year survival among colorectal cancer patients (C18–C20), stratified by the occurrence of a second primary malignancy and the use of hypolipidemic drugs, according to clinical stages. SPMs, second primary malignancies; HYP, hypolipidemic treatment; CRC, colorectal cancer.

**Table 1 cancers-14-01699-t001:** Characteristics of colorectal cancer patients (C18–C20), stratified by the occurrence of a second primary malignancy.

	No SPM (N = 1100)	With SPM (N = 301)	*p*-Value
**Gender**			
Men	680 (61.8%)	175 (58.1%)	0.257 ^1^
Women	420 (38.2%)	126 (41.9%)	
**Age at CRC diagnosis**			
0–44	90 (8.2%)	16 (5.3%)	**<0.001** ^1^
45–54	165 (15.0%)	25 (8.3%)	
55–64	350 (31.8%)	67 (22.3%)	
65–74	323 (29.4%)	125 (41.5%)	
75+	172 (15.6%)	68 (22.6%)	
Median (25–75% percentile)	63 (55–71%)	69 (61–74)	**<0.001** ^2^
**Clinical stage**			
Complete records	1 066 (96.9%)	287 (95.3%)	**0.014** ^1^
Stage I	273 (25.6%)	75 (26.1%)	
Stage II	266 (25.0%)	89 (31.0%)	
Stage III	304 (28.5%)	85 (29.6%)	
Stage IV	223 (20.9%)	38 (13.2%)	
Not available	34 (3.1%)	14 (4.7%)	
**Grade**			
Complete records	762 (69.3%)	241 (80.1%)	0.121 ^1^
1	190 (24.9%)	52 (21.6%)	
2	424 (55.6%)	152 (63.1%)	
3	148 (19.4%)	37 (15.4%)	
Not available	338 (30.7%)	60 (19.9%)	
**Relapse**			
yes	362 (32.9%)	64 (21.3%)	**<0.001** ^1^
no	738 (67.1%)	237 (78.7%)	

^1^ Fischer exact test, ^2^ Mann−Whitney test. SPM, second primary malignancy; CRC, colorectal cancer.

**Table 2 cancers-14-01699-t002:** Second primary malignancy in patients with colorectal cancer (C18–C20).

Patients with CRC	Men (N = 855)	Women (N = 546)	Total (N = 1401)
No SPM	680 (79.5%)	420 (76.9%)	1100 (78.5%)
With SPM	175 (20.5%)	126 (23.1%)	301 (21.5%)
One SPM	144 (16.8%)	102 (18.7%)	246 (17.6%)
Two SPMs	27 (3.2%)	20 (3.7%)	47 (3.4%)
Three SPMs	4 (0.5%)	4 (0.7%)	8 (0.6%)

SPM, second primary malignancy; CRC, colorectal cancer.

**Table 3 cancers-14-01699-t003:** The relationship between the use of hypolipidemic drugs and risk of a second primary malignancy in patients with colorectal cancer (C18–C20).

	No SPM	With SPM	*p*-Value
**Use of hypolipidemic drugs**			
No (N = 1144)	891 (77.9%)	253 (22.1%)	0.240
Yes (N = 257)	209 (81.3%)	48 (18.7%)	
**Use of statins ^1^**			
No (N = 1168)	909 (77.8%)	259 (22.2%)	0.093
Yes (N = 229)	190 (83.0%)	39 (17.0%)	
Lipophilic (N = 210)	178 (84.8%)	32 (15.2%)	**0.025**
Hydrophilic (N= 19)	12 (63.2%)	7 (36.8%)	
**Use of fibrates ^1^**			
No (N = 1370)	1079 (78.8%)	291 (21.2%)	0.634
Yes (N = 27)	20 (74.1%)	7 (25.9%)	

^1^ Patients taking ezetimibe were excluded (4 patients); SPM, second primary malignancy.

**Table 4 cancers-14-01699-t004:** Odds ratios for the occurrence of second primary malignancies derived from the multivariate logistic regression models.

	OR (95% CI)	*p*-Value
**Use of hypolipidemics**		
No	1.00	
Yes	0.74 (0.51–1.06)	0.099
**Use of statins ^1^**		
No	1.00	
Yes	0.62 (0.42–0.92)	**0.018**
Hydrophilic	1.00	
Lipophilic	2.80 (0.78–9.99)	0.329
**Use of fibrates ^1^**		
No	1.00	
Yes	1.71 (0.69–4.24)	0.251

^1^ Patients taking ezetimibe were excluded (4 patients).

**Table 5 cancers-14-01699-t005:** The relationship between second primary malignancies and laterality of colorectal cancer, stratified by the use of hypolipidemic drugs and excluding patients with C18.4 (transverse colon).

	No use of Hypolipidemics (N = 1091)			Use of Hypolipidemic Drugs (N = 248)			Total (N = 1339)		
	No SPM (N = 853)	With SPM (N = 238)	*p*-Value	No SPM (N = 202)	With SPM (N = 46)	*p*-Value	No SPM (N = 1055)	With SPM (N = 284)	*p*-Value
Right colon	151 (17.7%)	57 (23.9%)	**0.036**	43 (21.3%)	10 (21.7%)	0.736	194 (18.4%)	67 (23.6%)	**0.040**
Left colon	254 (29.8%)	76 (31.9%)		60 (29.7%)	16 (34.8%)		314 (29.8%)	92 (32.4%)	
Rectum	448 (52.5%)	105 (44.1%)		99 (49.0%)	20 (43.5%)		547 (51.8%)	125 (44.0%)	

SPM, second primary malignancy.

**Table 6 cancers-14-01699-t006:** Second primary malignancies by the site of diagnosis, stratified by the use of hypolipidemic drugs.

	No Use of Hypolipidemics (N = 304)	Use of Hypolipidemics (N = 53)	All Malignancies According to CNCR (N = 1,070,801)
Head and neck cancers (C00–C14, C32)	11 (3.6%)	3 (5.7%)	28,501 (2.7%)
Stomach (C16)	4 (1.3%)	4 (7.5%)	22,385 (2.1%)
Colon and rectum (C18–C20)	59 (19.4%)	14 (26.4%)	112,410 (10.5%)
Liver and intrahepatic bile ducts (C22)	7 (2.3%)	0 (0.0%)	12,500 (1.2%)
Pancreas (C25)	2 (0.7%)	0 (0.0%)	28,463 (2.7%)
Lung, bronchus, and trachea (C33, C34)	10 (3.3%)	3 (5.7%)	91,145 (8.5%)
Malignant melanoma of skin (C43)	13 (4.3%)	2 (3.8%)	29,507 (2.8%)
Other malignant neoplasms of skin (C44)	6 (2.0%)	1 (1.9%)	289,780 (27.1%)
Breast (C50)	55 (18.1%)	5 (9.4%)	92,356 (8.6%)
Cervix uteri (C53)	9 (3.0%)	0 (0.0%)	13,585 (1.3%)
Uterus (C54, C55)	7 (2.3%)	2 (3.8%)	26,677 (2.5%)
Ovary (C56)	5 (1.6%)	1 (1.9%)	15,482 (1.4%)
Prostate (C61)	36 (11.8%)	3 (5.7%)	84,720 (7.9%)
Testis (C62)	4 (1.3%)	0 (0.0%)	6614 (0.6%)
Kidney (C64)	27 (8.9%)	7 (13.2%)	41,511 (3.9%)
Bladder (C67)	11 (3.6%)	6 (11.3%)	30,948 (2.9%)
Thyroid gland (C73)	4 (1.3%)	0 (0.0%)	13,379 (1.2%)
Lymphomas (C81–C86)	9 (2.6%)	0 (0.0%)	22,847 (2.1%)
Leukemia (C91–C95)	6 (2.0%)	0 (0.0%)	19,041 (1.8%)
Other malignant neoplasms	19 (6.3%)	2 (3.8%)	88,950(8.3%)

Only SPMs with a known date of diagnosis were considered (date of diagnosis was not available for 7 SPMs). SPM, second primary malignancy; CNCR, Czech National Cancer Registry (2003–2016).

## Data Availability

The datasets generated and analyzed during the current study are available from the corresponding author on reasonable request.

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
