# Peer review of "Use of Hypolipidemic Drugs and the Risk of Second Primary Malignancy in Colorectal Cancer Patients"

_cancers, 2022, doi:10.3390/cancers14071699_

Round 1
Reviewer 1 Report
A well written and interesting article about hypolipidemic drugs and second primary cancer after primary colorectal cancer with a great interest to readers.
In the material and methods the exclusion criteria is not totally clear, maybe a table.
The primary endpint was 15-year survival but there are no numbers how many was followed for that time and how many patients that were left at 15 years. There is nothing about how many patients that was lost to follow up or excluded for other reasons.
The tables are hard to follow and table 4 must be more clarified to associate with the text. Table 5 is so much text so its hard to see the relevant facts. There is also numbers of colon and rectum cancer but it should be secondary primary cancers?
From figure 1 there is no explanation about the high number of skin neoplasms in the national registry compared to neoplasms in this material.
The unfavorable results of statins on early cancer is not so much discussed and could be of high significance in general practice.
Author Response
Please see the attachement.

Reviewer 2 Report
as file

Reviewer 3 Report
Halámková et al have critical described "Use of hypolipidemic drugs and the risk of second primary malignancy in colorectal cancer patients". This is a very important study for the field as there are several discussion on the impact of statin as potential adjuvant cancer therapy. For 20 past years there are several reports that using statins in long term could be a positive factor for patients to protect from cancer or have better response to chemotherapy when they have cancers. In current paper the respected team bring up the discussion that there might be a potential correlation between some specific cancers and use of statin.
1- Why the respected author choose colorectal cancer? It could be the same for cancer with KRAS mutation like lung cancer. I appreciate a detialed discussion on it in the discussion of the paper.
2- The respected author must do analysis considering hydrophilic and non-hydrophilic statin and then provide detailed funding for each of statin otherwise the result wont be reliable.
3- Statins are inducer of autophagy and UPR (unfolded protein response). It is very important the respected authors provide a deep discussion about these mechanisms and its potential impact in initiating secondary "primary" cancers in colorectal patients.
Round 2
Reviewer 3 Report
I appreciate the hard efforts of the respected authors. All of my concerns have been properly addressed.